# Incorporation of Powder Particles into an Impeller-Stirred Liquid Bath through Vortex Formation

**DOI:** 10.3390/ma14112710

**Published:** 2021-05-21

**Authors:** Sergey V. Komarov, Takuya Yamamoto, Hirotada Arai

**Affiliations:** 1Graduate School of Engineering, Tohoku University, Sendai 980-8579, Japan; takuya.yamamoto.e6@tohoku.ac.jp; 2Industrial Systems Engineering Department, Hachinohe National College of Technology, Hachinohe 039-1192, Japan; arai-c@hachinohe-ct.ac.jp

**Keywords:** particle incorporation mechanism, water model experiment, n simulation, surface turbulent fluctuation, capillary force

## Abstract

The present study addresses the incorporation of fine particles into liquids via the creation of a large-scale swirling vortex on the liquid free surface using a rotary impeller positioned along the axis of a cylindrical vessel. Four types of particles are used in the experiments to investigate the incorporation efficiency of the particles into a water bath under different impeller rotation speeds. Additionally, the vortex characteristics are investigated numerically. The results reveal that two factors, namely the particle wettability and turbulent oscillations at the bottom part of vortex surface, play dominant roles in determining the particle incorporation behavior. Hydrophobic particles are incapable of being incorporated into the water bath under any of the conditions examined in the present study. Partly wettable particles are entrained into the water bath, with the efficiency increasing with the impeller rotation speed and particle size. This is because an increase in the impeller rotation speed causes vortex deformation, whereby its bottom part approaches the impeller blades where the turbulent surface oscillations reach maximum amplitudes. Another possible mechanism of particle incorporation is the effect of capillary increases of liquid into the spaces between particles, which accumulate on the bottom surface of the vortex.

## 1. Introduction

Particle incorporation is the first stage of a process involving loading and dispersion of particles into a liquid. Incorporation and dispersion play key roles in such industrial applications as molten metal fluxing and refining, fabrication of metal matrix composites, mineral processing, and biochemical and pharmaceutical engineering. Recently, tremendous progress has been made in the development of highly efficient techniques to disperse fine particles in liquids, including molten metals. Although particle dispersion is not an easy task and many problems still remain unsolved or incompletely solved, the key factors influencing the dispersion efficiency are well known. As dispersion is a phenomenon occurring when particles are fully submerged into liquid, unsteady flow and turbulence are of primary importance [1]. Hence, high-intensity mechanical stirring and shock waves, including ultrasonic or hydrodynamic cavitation, have been long recognized as the most efficient tools for particle dispersion [2,3]. However, the phenomenon of particle incorporation into liquids has been investigated to a much lesser extent. Contact between particles and a liquid occurs firstly at the liquid’s surface, regardless of which way the particles are loaded onto the liquid. Therefore, capillary phenomena, particle wettability, liquid surface mobility, and turbulence can have determining effects on the particle incorporation. Nevertheless, in many studies devoted to the dynamics and dispersion of fine particles in liquids, particle incorporation is not considered at all or is oversimplified.

The fact that wettability and surface phenomena govern the particle behavior at the gas–liquid interphase is well known. The influence of the surface phenomena is especially great when micro-sized poorly wettable particles are incorporated into a liquid [4]. This problem is of prime significance for molten metals, as many powdered materials have poor wettability with most molten metals and are often lighter than metal. This is why many attempts have been made to improve the incorporation efficiency of particles into molten metals. For example, the use of the kinetic energy of particles to inject them into molten metals with a carrier gas is potentially effective approach for particle incorporation; however, gas injection produces bubbles in molten metals, causing particle flotation [5,6]. Thus, the application of this method is practically limited to relatively coarse particles. Another promising technique is to load particles in a compacted form, for example as briquettes or pellets. This technique, however, has a significant limitation in that such compacted samples need to be fragmented and dispersed in the liquid, which is not easy. Furthermore, in liquid metals, compacted particles may be sintered due to high temperatures, which makes their dispersion even more difficult. This is why loading of particles onto the free surfaces of liquids followed by liquid agitation or ultrasonic treatment is considered one of the more efficient techniques for particle dispersion. When using a combination of ultrasound and agitation, ultrasonic irradiation improves the wettability of particles by liquid, which makes it possible to incorporate non- or poorly wettable particles. For example, Genma et al. [7] showed that micron-sized alumina particles can be incorporated into a mechanically stirred aluminum melt irradiated with ultrasound. Simitomo et al. [8] investigated the dispersion of polymethylmethacrylate (PMMA) particles in aqueous solutions using both mechanical stirring and ultrasonic irradiation. They found that the dispersion efficiency of ultrasonic irradiation is larger than that of the mechanical stirring at lower input power, however the situation reverses at larger input powers.

The charging of powder particles onto molten metal surfaces followed by ultrasonic irradiation, as demonstrated by Yang et al. [9], is another possible way to improve the particle incorporation efficiency. The authors used SiC particles to fabricate an aluminum alloy A356 base composite material, then characterized its microstructure and its ability to exploit such phenomena as acoustic streaming and cavitation, which probably cause turbulent and oscillatory motions on the molten metal’s free surfaces. Du et al. [10] performed ultrasonic treatment of an Al_2_O_3_-particle-containing aluminum melt sequentially in two steps at two different frequencies, first at 10 and then at 20 kHz. The Al_2_O_3_ particles added onto the melt surface contained a certain amount of Mg, which allowed them to improve the wettability of alumina particles in the melt. Although this method provides an attractive approach to particle incorporation, the high temperatures and large volume of molten metal impose serious limitations on the use of this technique.

On the other hand, the so-called classical method, in which impellers are used to agitate the melt and create a vortex on the melt surface, offers more flexibility in terms of treatment cost and productivity. In this method, powder particles are loaded first onto the vortex surface, followed by particle incorporation and dispersion. Sukawa et al. [11] investigated the dispersion of fine particles (diameter of 30–50 μm) into molten metals using a water bath stirred by an impeller. They found that the particle dispersion in an impeller-stirred water bath can be significantly improved by immersing a cylinder in the bath at an offset radial position. High shear mechanical agitation provides highly efficient dispersion of particles in the melt; however, the low efficiency of the particle incorporation is one of the main shortcomings of this method. Although the mechanism underlying particle incorporation is not properly understood yet, it is obvious that both the interface properties and unsteady flow phenomena occurring near and at the vortex’s free surface significantly affect the incorporation efficiency. Below is a brief survey of the recent studies relating to the interface phenomena and behavior of solid particles at the gas–liquid interface, with an emphasis on unsteady flow phenomena.

It is noted that the great majority of studies have focused on the behavior of single particles at a stationary gas–liquid interface. The results have revealed that when a particle rests on a smooth steady surface of a liquid, it floats on the surface, remaining partially submerged in the liquid to a depth that is dependent on the particle size, density, and wettability from the liquid [12]. Innes-Gold et al. [13] found that the behavior of single particles on the interface between two liquids can be described by the following three parameters: the contact angle, the Bond number, and the ratio of the particle’s buoyant density to liquid-phase densities. The experiments and numerical simulation performed by Liu et al. [14] revealed that the capillary force can prevent a solid particle from penetrating when it falls onto a liquid surface, even if the particle is hydrophilic and has a greater density than the liquid. This tendency is especially apparent for nano- and micro-sized particles. Additionally, a particle’s shape can affect the depth of its submergence into liquid [15]. It should be noted, however, that the shape effect is not of crucial importance for fine particles, because their shape should be close to spherical. Several studies have been devoted to observation or prediction of the behavior of two or more particles on gas–liquid surfaces. It is interesting to note that interactions between floating particles can be both attractive and repulsive. At the moment a particle comes into contact with a gas–liquid surface, it starts to oscillate, generating a capillary wave at the gas–liquid surface. As a result, the particles repel each other and become dispersed. This phenomenon was explained in a review paper by Singh et al. [12]. The particle oscillation originates from the capillary force, which pulls particles into the interface, causing them to accelerate. As the motion of the particles is inertia-dominated, they oscillate about the equilibrium position with gradually decreasing amplitude due to viscous drag and eventually come to a stop. After this, the particles can remain dispersed, or if an attractive force pulls them together, they tend to cluster. The latter tendency can be observed in both the hydrophilic and hydrophobic cases depending on the particle density and size.

The behavior of single particles on oscillating or wavy gas–liquid interfaces has also been investigated in the past, mainly in the field of particle bubble flotation. Nevertheless, the results of these studies are of interest regarding the particle incorporation considered in the present study. Stevenson et al. [16] investigated forces acting on a particle attached to the meniscus of an oscillating liquid–gas interface. Their approach takes into account not only commonly known capillary and buoyancy forces, but also the inertia force, the Basset history force, the added mass of particles, and the drag force imparted by the liquid upon the particles, and therefore allows a more accurate prediction of particle behavior. At a constant particle size, the effects of oscillation were shown to become stronger with increasing oscillation frequency. On the whole, in order to enhance the entrainment of particles measuring several microns in diameter into a liquid from a gas–liquid interface, the oscillating frequency should be much higher than 100 Hz. Bochkarev et.al [17] concluded that detachment of even a large particle from the free surface of a liquid becomes possible when the oscillating frequency is as high as several hundred Hz. In multiparticle systems, free surface turbulence has a determining effect on the particles floating on the surface. Pratt et al. [18] showed that Lagrangian coherent structures generated by a turbulent flow near the gas–liquid free surface cause clustering of the initially well-mixed buoyant particles floating on it. The results of another study [19] suggested the same phenomena. Moreover, these authors have shown that the clustering effects connected to the formation of sources and sinks of fluid velocity are generated by subsurface upwelling and downwelling motions under the liquid’s free surface turbulence.

The above-mentioned phenomena have a direct bearing on the incorporation of particles from the vortex’s free surface into a liquid bath stirred by impellers. Certain features of surface vortex formation and free surface turbulence have been investigated in our previous studies. In particular, Yamamoto et al. [20] showed that on-axis impeller stirring leads to formation of not only a large vortex around the impeller axis, but also trailing vortices, which significantly affect the turbulent motion of the free surface of the liquid. The goal of the present study is to investigate the incorporation and dispersion of particles into a water bath from a free surface vortex, and thus to clarify the key variables and mechanisms influencing the efficiency of the vortex treatment. Partly wettable and non-wettable particles are used to bring the conditions closer to those typical for molten metals. The study includes three parts, namely an experimental investigation using a water model, numerical simulation of the vortex formation and its characteristics, and discussion of the results obtained.

## 2. Experiments

### 2.1. Setup and Measurement Procedures

Figure 1 presents a schematic drawing of the experimental setup. A transparent acrylic cylindrical vessel with an inner diameter of 94 mm and height of 200 mm was used for the experiments. The vessel was filled with about 1.2 L of water. Thus, the depth of the water bath in the vessel was about 168 mm. The vessel was equipped with two 6 mm sampling holes to take samples of particle-containing water during experiments. An impeller with 4 blades measuring 10 mm in width was used to agitate the water bath at clockwise rotation speeds of 250, 450, and 650 rpm. The blades were inclined at 45 degrees to the horizontal plane.

The impeller was positioned vertically at the vessel axis in such a way that the distance between its blades’ lower edges and the vessel bottom was 80 mm. The impeller diameter (distance between the opposite tips of impeller blades) was 50 mm.

The experimental procedure was as follows. The impeller stirring was accompanied by the formation of a vortex around the impeller shaft, as shown in Figure 1. The vortex depth was dependent on the impeller rotation speed. More details on the vortex shape will be given in the discussion section. After agitation of water bath for 2 min to stabilize the water flow in the vessel, a fixed amount of particles was added onto the vortex surface. In the majority of experiments, the particle amount was 2.5 g. The particle size and other characteristics are summarized in Table 1. These data were provided by the particle suppliers, which are also indicated in Table 1. Additionally, the particle size was measured using a Beckman coulter particle size analyzer (Multisizer 3). The particle size was verified by SEM observations. Figure 2 and Figure 3 present the typical appearances of acrylic and PTFE particles and their distributions in size. The wetting angles were measured using a novel technique proposed earlier by one of the authors of the present study on liquid–particle measurements. The details are given in Appendix A.

As readily seen in Table 1, the wetting angles vary from 76° to 157°, showing partial wetting of PMMA and acrylic particles, and nonwetting of PTFE particles. For simplicity, hereinafter these particles will be referred to as hydrophilic and hydrophobic.

Samples were taken using the above two tubes at 60, 180, 300, and 600 s after adding the particles. Samples were diluted with a predetermined amount of water and introduced into the measurement chamber of the above-mentioned Beckman–Coulter particle size analyzer to measure the particle frequency distribution in diameter. Then, the measured data were inserted into Equation (1) to determine the ratio of particle incorporation, *α*:(1)$$\alpha =\frac{\rho \sum_{i}^{n}N_{i}V_{i}}{M_{0}}$$
where *ρ* is the particle density, *N_i_* is the particle number in each diameter range, *V_i_* is the particle volume, and *M*_0_ is the weight of added particles.

### 2.2. Experimental Results

Using the above experimental setup and procedure, the kinetics of particle incorporation were investigated. Figure 4a–c shows time variations of the incorporation ratio measured with particles A and B. A number of features can be seen in this figure. For all three conditions, the incorporation ratio is increased with stirring time. However, the increases vary depending on the rotation speed and particle size. In the experiments with particle B, *α* increases up to 1 for a relatively short time at a rate independent of the rotation speed.

The value of *α* was kept at around 1 or decreased down to 0.7–0.8. The latter was presumably because a part of particles sunk down to the vessel bottom. The A particles, which were larger, were entrained into the water bath slower compared to the B particles, reaching a state of full incorporation after stirring for 600 s. Notably, in the first half of experiments with the larger A particles, the higher rotation speed of impeller led to a faster increase in the incorporation ratio during stirring. The same tendency was observed for the smaller A particles. As can be seen from Figure 4c, the incorporation ratio *α* increased as the rotation speed became higher. However, even at the highest rotation speed and longest stirring time examined in these experiments, the value of *α* did not exceed 0.6, which was smaller than for the larger A particles.

The incorporation ratio of hydrophobic C particles was found to be close to 0 for all experimental conditions examined. Therefore, these data are not presented graphically in the figures. Thus, this result suggests that the contact angle has a dominant influence on the particle behavior during the incorporation experiments.

In an attempt to understand the particle incorporation mechanism and to elucidate which factors influence the particle behavior, the stirring process was simulated numerically. The next section explains the simulation details and presents the computed results.

### 2.3. Numerical Simulation Procedure and Results

The simulation technique was the same as that used in our previous study. Hence, the details, including the model validation, can be found in our earlier papers [20,21]. Briefly, an open source software, OpenFOAM (Version 2.3x, Software for CFD; OpenCFD Ltd.: Bracknell, UK), was applied for the calculations. The solver used was interDymFoam, which can solve the liquid–gas multiphase flow with dynamic mesh motion. For the interface capture, the algebraic volume of fluid (VOF) method [22] was used in the solver. The impeller rotation was set in the model by using the sliding grid technique. The numerical grid was rotated in the inner part of the numerical domain and was stationary in the outer part of the numerical domain. The interface between rotating and stationary parts was connected by the arbitral mesh interface (AMI) boundary condition. The total number of grid cells was about 1,130,000 cells. The size and shape of the vessel and impeller were similar to those used in the experiments.

The main goal of the numerical simulation in this study was to predict the vortex shape and turbulent characteristics at the vortex surface. In order to avoid complexity and achieve computational efficiency, it was assumed that the particles had no influence on the vortex characteristics. This is because the weight of added particles was only 0.2% of that of the water bath. Additionally, visual observations revealed that the behavior of the vortex before and after particle addition remained unchanged. Thus, the particles were excluded from the simulation and only the behavior of the fluid phase was simulated. Figure 5 presents the time-averaged free surface shapes for all three rotation speeds examined in this study. It can be seen that the free surface was deformed, descending gradually toward the impeller blades with increasing rotation speed, reaching the blades at 650 rpm. The same tendency was observed experimentally. This suggests that the present model is valid to predict the vortex shape. Additionally, these data clarify why 650 rpm is the maximum rotation speed examined in the present study. Any further increase in the rotation speed will result in entrainment of air from the bottom surface of the vortex into the water bath, which is unacceptable for such experiments.

In order to evaluate the turbulent characteristics of the vortex, its oscillation amplitude was calculated first. The amplitude was assessed numerically in the following way. The time-averaged volume fraction of the liquid, *α_av_*, can be calculated as:(2)$$\alpha_{av}=\frac{\int_{t_{0}}^{t_{1}}\alpha dt}{t_{1}-t_{0}}$$
where *t*_1_ is the time-average end time and *t*_0_ is the time-averaged start time. The deviated volume fraction *α*′ can be expressed as:(3)$$\alpha^{\prime }=\alpha -\alpha_{av}$$

The deviated volume fraction of liquid is squared and time-averaged as:(4)$$\alpha_{sq,av}^{\prime }=\frac{\int_{t_{0}}^{t_{1}}\alpha^{\prime }{}^{2}dt}{t_{1}-t_{0}}$$

In the present study, *t*_0_ and *t*_1_ were set to 10 and 15 s, respectively. The squared and time-averaged deviation of the volume fraction, $\alpha_{sq,av}^{\prime }$, was used to evaluate the oscillation amplitude of the vortex surface. Figure 6 shows the predicted values of $\alpha_{sq,av}^{\prime }$ for each rotation speed. At least two features can be mentioned from these predictions. First, the amplitude becomes larger at the bottom locations of the vortex surface, regardless of the rotation speed. Second, the maximum amplitude is reached at the bottom location of the vortex surface under highest rotation speed of 650 rpm.

Next, the vortex surface’s fluctuating velocity was quantified from the results of the numerical simulation. Figure 7 presents the distributions of the mean square velocity $\bar{u{\;}_{z}^{\prime }u_{z}^{\prime }}$ over water volume near the vortex surface. Only the vertical z-components are presented because vertical fluctuating velocity mainly affects the particle incorporation. It can be seen that the fluctuating velocity reaches maximum values at the bottom part of vortex close to the rotation impeller. Based on these data, the root square mean (rms) velocity $u_{z}^{\prime }$ was calculated. Typical rms velocities at the bottom locations of the vortex surface were estimated to be 0.06, 0.12, and 0.24 m/s for rotation speeds of 250, 450, and 650 rpm, respectively. Then, one can calculate the frequency at which the vortex frequency oscillates as $u_{z}^{\prime }/A$, where A is the oscillation amplitude, which can be obtained from Figure 7. For example, a typical value of A for 650 rpm can be estimated to be approximately 5 mm from Figure 7c for the area indicated by the white arrow. Therefore, the vortex oscillating frequency f_v_ is equal to 48 Hz. It is interesting to note that this value is about one-quarter of that of the impeller rotation, which is equal to 650/60 = 10.8 Hz for the considered case. This finding suggests that the vortex surface oscillations are caused by passage of each of the four impeller blades, and accordingly the oscillating frequency can be calculated to be 10.8 × 4 = 43.2 Hz. This value is quite close to the above frequency of 48 Hz.

## 3. Discussion

Based on the above experimental results and numerical predictions, possible mechanisms of particle incorporation will be discussed in this section. As follows from the above experimental results, the contact angle and capillary forces have a dominant influence on the particle incorporation. Figure 8a is a schematic drawing showing the forces acting on a particle floating on the vortex surface. They include the capillary force F_s_, buoyancy force F_b_, gravitational force F_g_, and centrifugal force F_c_ arising due to the vortex surface rotation. The particle entrainment depth is denoted by h. It is notable that the capillary force vector may be directed not necessarily outward from the vortex interface as shown in Figure 8a, but also toward it depending on the contact angle. Figure 8b–e shows the contact interface geometry and contact angles for four typical situations: hydrophilic particles *θ* < 90° (b,d) and hydrophobic particles *θ* > 90° (c,e) for small (b,c) and large (d,e) entrainment depths of a particle in liquid. For convenience of consideration, the gas–liquid interface is depicted horizontally. Fs represents the magnitude of vertical component of the capillary force vector. It can be seen that when the particle surface is poorly wetted by liquid, the vector of capillary force is always directed outward from the free surface. This is the main reason for the very low incorporation efficiency of particles with poor wettability. The behavior of sufficiently wetted particles is quite different. At small depths of particle submersion, the capillary force is directed toward the liquid surface, contributing to the particle entrainment into the liquid, as shown in Figure 8b. However, when the submersion depth exceeds the particle radius, the direction of the capillary force is turned upward, which prevents the particles from entrainment into the liquid, as seen from Figure 8d. This is why even hydrophilic particles, having relatively good wettability, can be held on the liquid surface by the capillary force.

Each of the above forces can be expressed in term of the particle radius, Rp, and appropriate physical properties. For example, force components normal to the vortex surface can be expressed by the following equations [16]:(5)$$F_{g}=\frac{4}{3}\pi R_{p}^{3}\rho_{p}g\cos\varphi$$
(6)$$F_{b}=\frac{\pi }{3}R_{p}^{3}\rho_{l}g\left(2+3\cos\alpha -\cos^{2}\alpha \right)\cos\varphi$$
(7)$$F_{s}=-2\pi R_{p}\sigma \sin\alpha \sin\left(\alpha +\theta \right)$$
(8)$$F_{c}=\frac{4}{3}\pi R_{p}^{3}\rho_{p}\omega^{2}L\sin\varphi$$
where *φ* = 90° − *β*; *β* is the angle between the vertical line and vortex’s free surface (Figure 8a); *θ* is the contact angle, *α* is the angle between the horizontal line and line connected to the particle center and three-phase contact point as indicated in Figure 8d; *ρ_p_* and *ρ_l_* are the densities of particle and liquid, respectively; *σ* is the liquid surface tension; *ω* is the angular velocity of impeller rotation; *L* is the distance between the rotation axis and vortex surface, as shown in Figure 8a.

A comparison of order-of-magnitude estimates for these forces is shown in Table 2 for the three representative types of particles, A, B, and C (Table 2). It can be seen that the ratios for the three forces *F_g_*, *F_b_*, and *F_c_* to capillary force *F_s_* are much smaller than unit, suggesting that the behavior of particles is fully determined by the capillary forces, independent of the particle sizes and properties. In these estimates, all trigonometrical functions were considered to equal the unit. The surface tension of the water was set to be 0.072 N/m. To estimate the centrifugal force, Fc, the highest angular frequency of *ω* = 68 rad/s and distance of L = 0.3 m were used. This distance corresponds to the vortex radius at the water-free surface under the above angular frequency. This condition corresponds to the maximum centrifugal force that can be created in the present experimental conditions. Thus, one can conclude that the centrifugal force is incapable of enhancing the particle incorporation efficiency.

Another factor that should be considered is the possible entrainment of particles into the oscillating motion of the vortex surface. Obviously, the oscillations may influence the particle incorporation into the liquid bath. This possibility can be analyzed by using the Stokes number, *St*, which is commonly used to characterize the dynamic behavior of particles suspended in a fluid flow. The Stokes number for spherical particles of diameter *d_p_* is defined as follows:(9)$$St=\frac{u_{0}t_{0}}{d_{p}}$$
where *u*_0_ is the fluid velocity of the flow away from the particle and *t*_0_ is the relaxation time of the particle. In the case of Stokes flow (the particle’s Reynolds number is less than unity), the relaxation time of the particle can be written as:(10)$$t_{0}=\frac{\rho_{p}d_{p}^{2}}{18\mu_{f}}$$
where *μ_f_* is the dynamic viscosity of fluid.

The calculated relaxation times and Stokes numbers for the A and B particles are presented in Table 2. The smallest particles exhibit a Stokes number smaller than 1. This means that these particles completely follow the turbulent eddy motion at the vortex surface in the present experiments. As for larger particles, the corresponding Stokes numbers are slightly larger than 1, suggesting that the inertia of the particles can play a role in their behavior. This may result in a phase shift between oscillations of particles and the vortex’s free surface where the particle is attached, forcing the particle to submerge deeper in the liquid during certain oscillation phases. This difference in the submergence depth at the stationary and oscillatory free surfaces, although small, would be sufficient for particles to be fully submerged into the liquid, being then entrained by turbulent eddies into the liquid bulk. This mechanism is most likely to occur in the case of hydrophilic particles of larger diameter, which are immersed into the liquid to a depth larger than their diameter (Figure 8c,d). On the other hand, hydrophobic particles, although they are big (particle C, Table 2), are unlikely to be entrained into the liquid according to this mechanism because their submergence depth at the free surface in the stationary condition is much smaller than to their diameter (Figure 8c).

Another possible mechanism of particle penetration is related to the liquid penetration due to the capillary increase. This phenomenon has been widely investigated in the past, particularly regarding packed beds. Most of the recent results were reviewed by Alghunaim et al. [23]. A packed bed, composed of fine particles, is very similar to a porous body, being permeable to liquid due to capillary forces. As the impeller rotation causes formation of a concave vortex on the surface, particles accumulate at the vortex bottom part forming a particle bed, which oscillates together with the vortex’s free surface. Porosity (or voidage) of such a particle bed should depend on the oscillation phase due to the packed bed inertia. More specifically, the porosity is expected to be lower when the vortex bottom surface begins to move upward, and vice versa higher when the surface begins to move downward. This is schematically illustrated in Figure 9. The first situation (Figure 9a) is assumed to be more favorable for the liquid penetration than the second one (Figure 9b), because in this case, in parallel with the capillary force, the inertial driving force also contributes to the liquid penetration. The penetration rate can be estimated quantitatively using well known theoretical and empirical expressions. The vertical height h_c_ of the liquid increase in a capillary within a bed of particles during time t can be calculated according to the following expression [24]:(11)$$\frac{dh_{c}}{dt}=\frac{9\left(1-\varepsilon \right)r_{C}\sigma \cos\theta }{75\mu h_{c}}$$
where *ε* is the particle layer porosity, *r_c_* is the capillary radius, *σ* is the surface tension, *θ* is the contact angle, and *μ* is the dynamic viscosity of the liquid.

Notice that Equation (11) was obtained from Equation (A5) (see Appendix A) for the case where hydrostatic pressure equals 0. As mentioned above, the layer porosity *ε* is assumed to vary depending on the vortex dynamic behavior. Hence, estimates were made for the lowest porosity of a bed composed of spherical particles of the same size. In this case, the value of *ε* can be taken as 0.375. The evaluated values of *h_c_* for hydrophilic particles A and B are plotted in Figure 10 as a function of time, t. For the sake of a better understanding of the time scale of the increase of the liquid, time in Figure 10 is presented in a dimensionless form obtained through division by the period of vortex surface oscillations, T_i_, determined from the predicted frequency of vortex surface oscillations as explained above. The results reveal that the height of the liquid increase in such a “capillary” can rapidly reach values significantly exceeding the particle diameters. Obviously, in the case of hydrophobic C particles, the liquid increase height is 0 because cos*θ* is negative. As seen from Figure 10, the liquid increase rate is significantly increased with the particle diameter. This could be one more reason why the larger particles penetrate into the liquid bath easier than fine particles. According to this mechanism, liquid increases in the capillary-like channels formed by the neighboring particles, and when the increasing height exceeds the particle diameter, the capillary force disappears and the particle is fully entrained by the liquid turbulent flow.

## 4. Conclusions

In this paper, experiments and simulations were carried out to investigate possible mechanisms of entrainment of fine particles charged onto the free surface of a vortex formed during mechanical stirring of a liquid bath by impellers. The following conclusions can be made from the above results.

Hydrophilic particles were entrained into the stirred liquid bath, with the efficiency increasing with the impeller rotation speed and particle size. Alternatively, hydrophobic particles did not penetrate the liquid bath under any conditions examined in the present study;Capillary force plays a dominant role in the behavior of particles on the vortex’s surface, while the effects of centrifugal force are insignificant under the present conditions;Increases in impeller rotation speed result in increases of the vortex depth, causing the accumulation of particles on the bottom part of the vortex surface near the impeller shaft, where the intensity of turbulent surface fluctuations reaches maximum values. This is considered as the main reason for particle incorporation in impeller-stirred vessels;Another possible mechanism of particle incorporation is the effects of capillary increases of liquid into the spaces between particles that have accumulated on the vortex’s bottom surface. This mechanism is more likely to occur at higher speeds of impeller rotation when a relatively thick layer of particles forms on the vortex surface.

## Figures and Tables

**Figure 1 materials-14-02710-f001:**
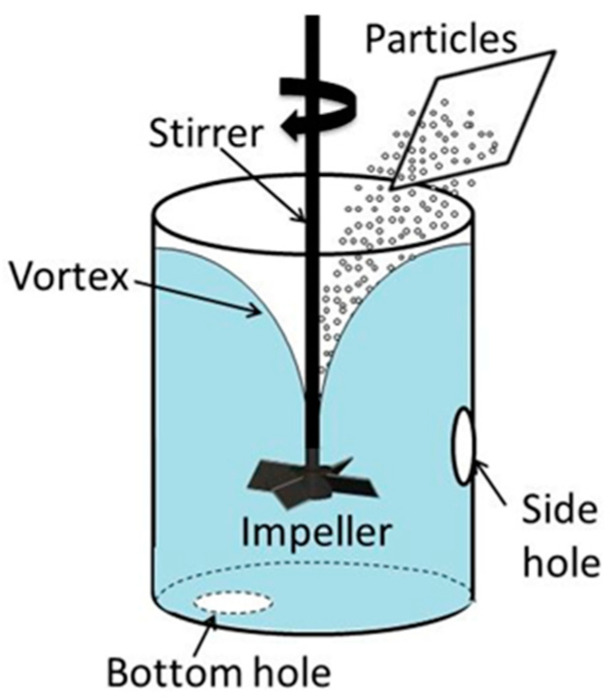
A schematic representation of the experimental setup.

**Figure 2 materials-14-02710-f002:**
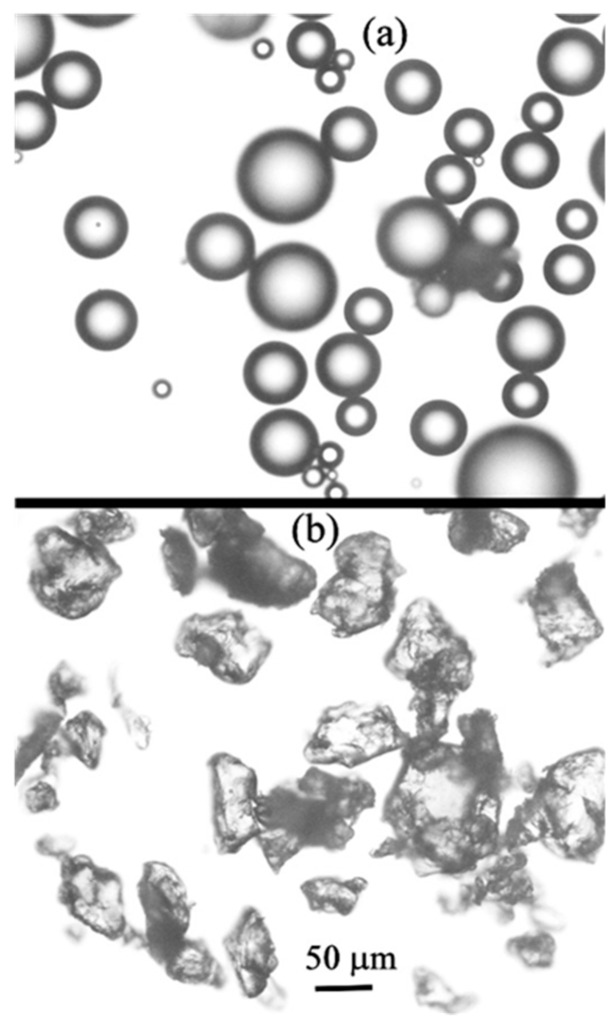
Appearances of acrylic (**a**) and PTFE (**b**) particles.

**Figure 3 materials-14-02710-f003:**
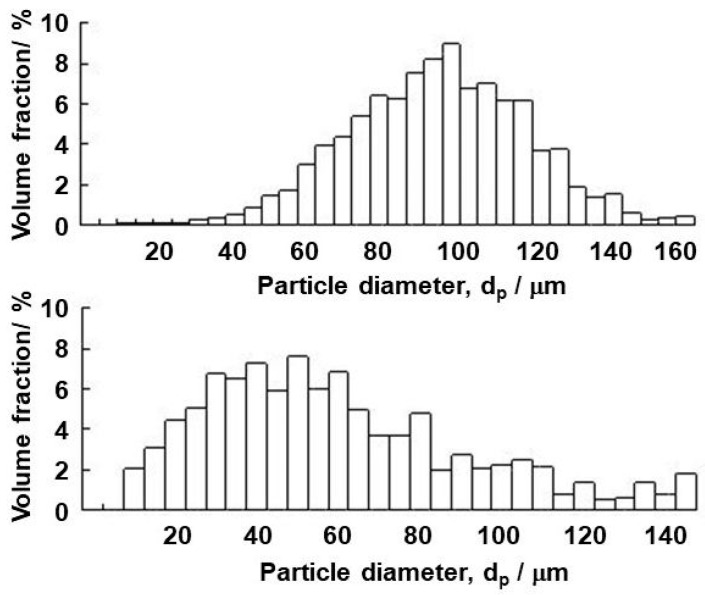
Size distributions of acrylic (**upper**) and PTFE (**bottom**) particles.

**Figure 4 materials-14-02710-f004:**
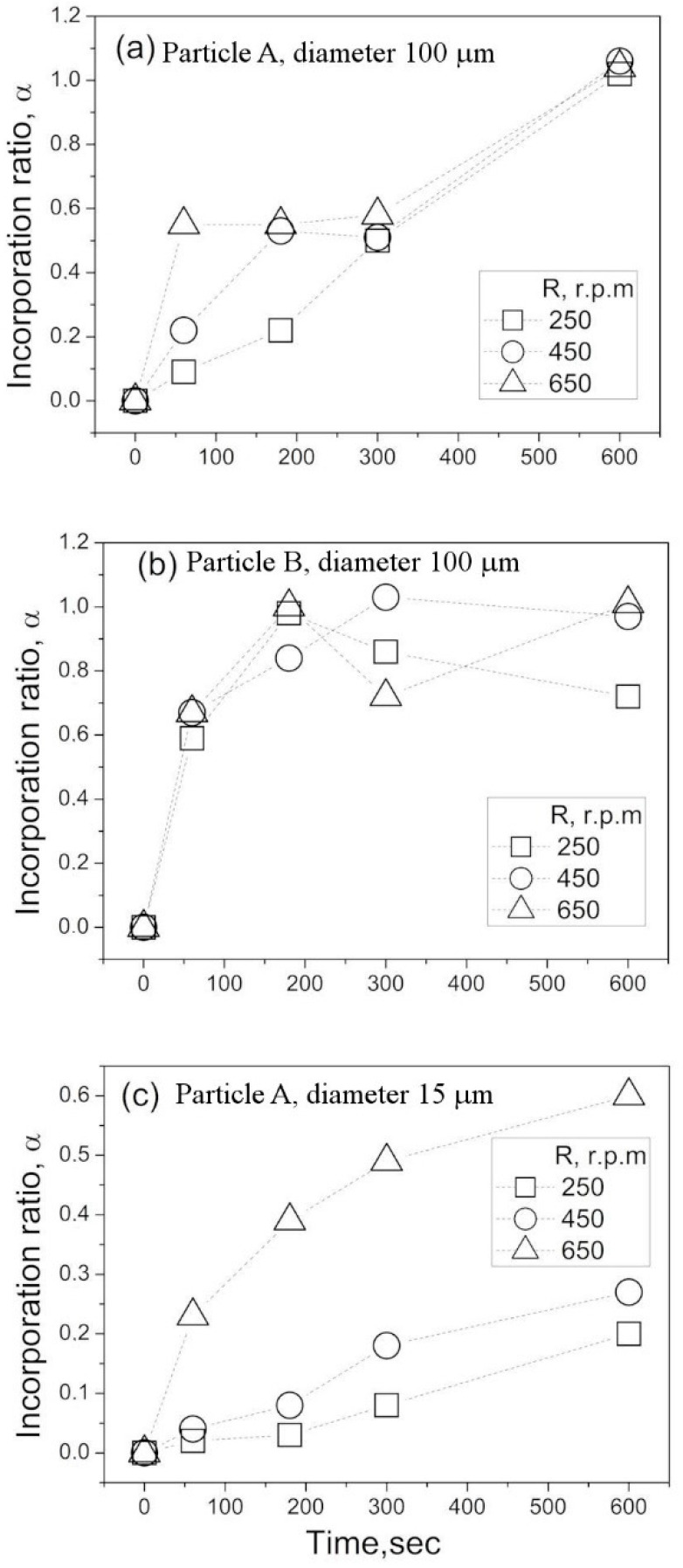
Time variation of particle incorporation ratio. (**a**) Particle A, diameter 100 μm, (**b**) Particle B, diameter 100 μm, (**c**) Particle A, diameter 15 μm.

**Figure 5 materials-14-02710-f005:**
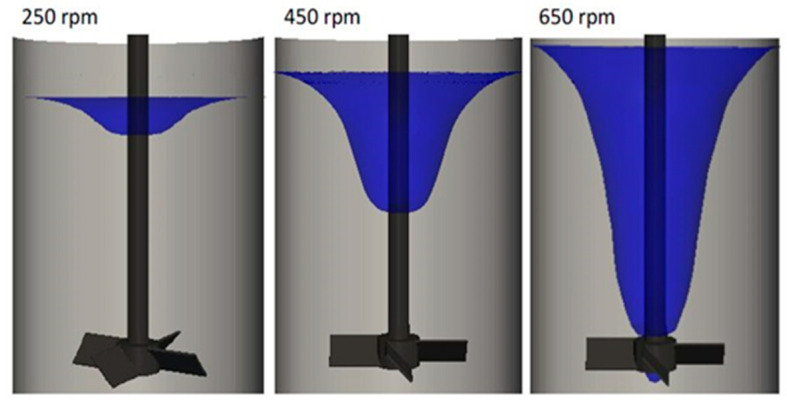
Predicted shapes of the large-scale vortex at different rotation speeds.

**Figure 6 materials-14-02710-f006:**
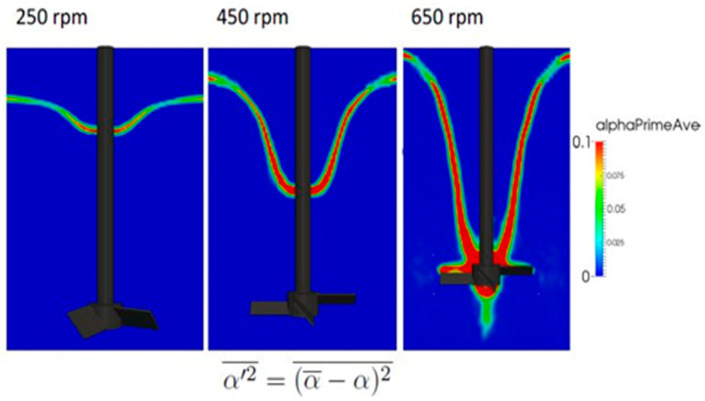
Predicted amplitudes of vortex surface oscillations.

**Figure 7 materials-14-02710-f007:**
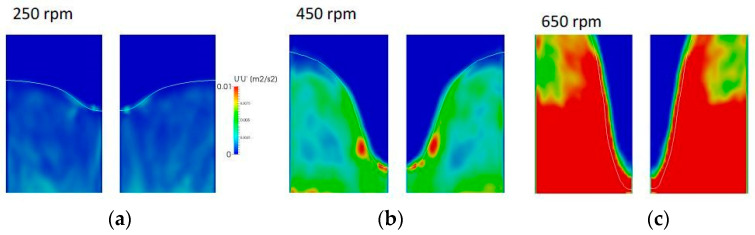
Predicted distribution of the mean square velocity in water near the vortex surface. (**a**) 250 rpm, (**b**) 450 rpm, (**c**) 650 rpm.

**Figure 8 materials-14-02710-f008:**
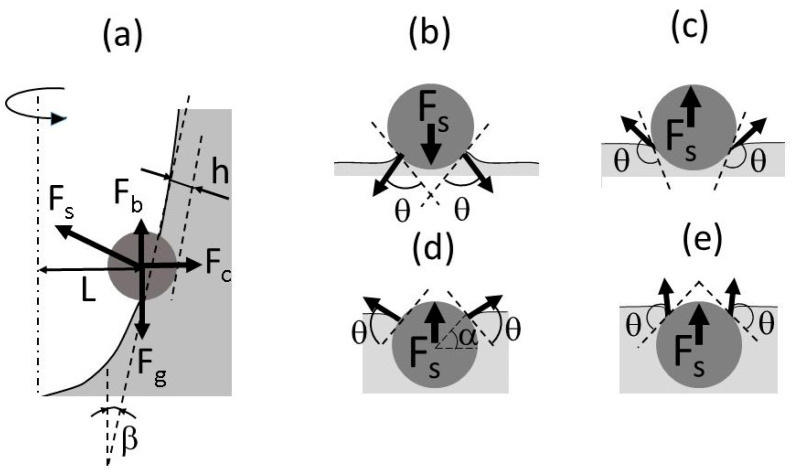
Forces acting on a particle trapped at the vortex surface (**a**) and contact angles of hydrophilic (**b**,**d**) and hydrophobic (**c**,**e**) particles.

**Figure 9 materials-14-02710-f009:**
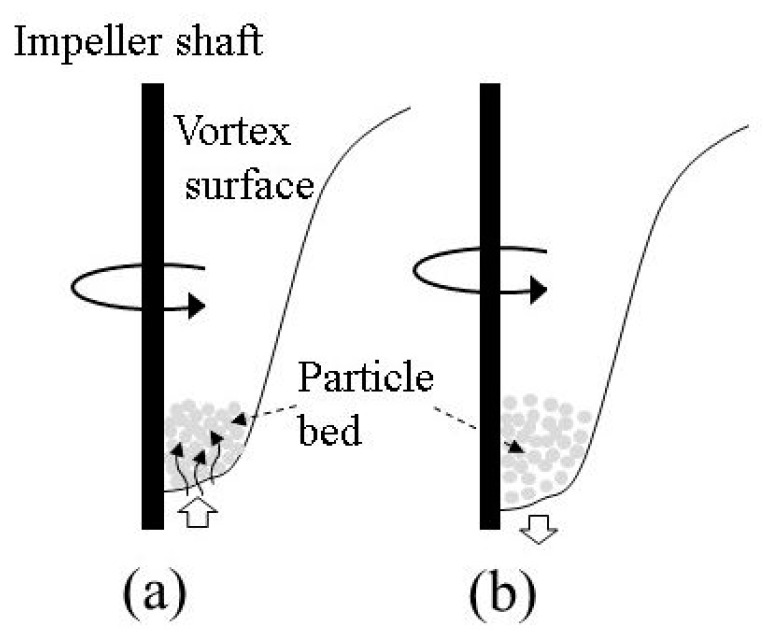
A schematic representation of the particle behavior at the bottom of the vortex during the surface oscillations. The surface moves (**a**) upward and (**b**) downward.

**Figure 10 materials-14-02710-f010:**
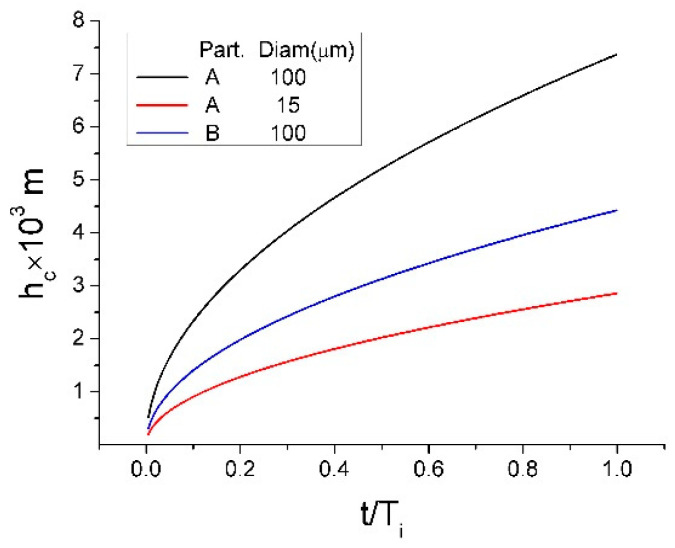
Time variation of the height increase of the liquid in the particle bed.

**Table 1 materials-14-02710-t001:** Properties of particles used in the experiments.

Particle	Material	Density, ρ_p_	Wetting Angle, *θ*	Diameter *, d_p_
		kg/m^3^	deg.	μm
A	PMMA	1200	76	15,100
B	Acrylic	1210	85	100
C	PTFE	2150	157	50

* Diameter corresponding to the maximum value of the particle size distribution.

**Table 2 materials-14-02710-t002:** Ratios of main forces acting on particles and their Stokes numbers.

Particle	Diameter	Density	*F_g_*/*F_s_* × 10^6^	*F_b_*/*F_s_* × 10^6^	*F_c_*/*F_s_* × 10^2^	*t*_0_ × 10^5^	*St*
	μm	kg/m^3^				s	
A	15	1200	6.1	5.1	0.087	1.43	0.23
	100	1200	273	227	3.86	63.4	1.52
C	50	2150	122	56.8	1.73	28.4	1.36

## Data Availability

Data sharing is not applicable to this article.

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
