# Peer review of "Incorporation of Powder Particles into an Impeller-Stirred Liquid Bath through Vortex Formation"

_materials, 2021, doi:10.3390/ma14112710_

Round 1

Reviewer 1 Report

The general organization of the paper and the clarity of the writing and content are good. Good explanation of experimental test, clear description of the numerical simulation. This paper can be accepted, I support it for publication.

  1. Is there any validation or comparison between experiments and numerical simulation?
  2. Which turbulence used in this paper? Did you consider the boundary layer mesh, what about the yplus? Does it meet the requirements of the used turbulence model?

Author Response

Please find our response in the file attached

Reviewer 2 Report

In the introduction, there should be more references to the literature in which areas this process occurs.

The description of the axis in Figure 3 should be more visible.

Line 186 should be instead of I - 1

Figure 4 should be larger and the symbols in the legend should be assigned the same value, i.e. triangle - 250, square - 650, because it makes the analysis of the chart difficult.

The calculation results presented in Figure 6 should be more detailed, the numerical values corresponding to the color scale are not visible.

Line 336 should be Table 2

Figure 10 should be corrected

Line 433 - Figure 1A should be Figure A1 

There are many editorial mistakes that need to be corrected

Author Response

Dear Reviewer,

We greatly appreciate the Reviewer’s valuable comments and suggestions concerning our paper. Please find below a list of our answers and changes to the manuscript.

Yours faithfully,

The corresponding author

Sergey Komarov

 In the introduction, there should be more references to the literature in which areas this process occurs.

We added 6 more references to the introduction section.

 The description of the axis in Figure 3 should be more visible.

Fig.3 is replaced by a new more visible figure.

 Line 186 should be instead of I - 1

The line is corrected

Figure 4 should be larger and the symbols in the legend should be assigned the same value, i.e. triangle - 250, square - 650, because it makes the analysis of the chart difficult.

The figure and symbols have been corrected according to the Reviewer’s comments.

Line 336 should be Table 2

The number of table has been corrected.

Figure 10 should be corrected

Description of X-axis in this figure was hidden by the caption. Now, the description is in the right position. Thank you for your comment.

Line 433 - Figure 1A should be Figure A1

1A is replaced by A1.

There are many editorial mistakes that need to be corrected

We apologize for so many mistakes in our manuscript. It is our belief that all mistakes indicated have been corrected. 

Reviewer 3 Report

I have read the paper with much interest given the potential applications of the results. The paper is well written and the results are clearly described.

I have only a comment:

Although the numerical simulations were quite adequate to help explaining the experiments, the disperse phase (i.e., the penetrating particles) is not accounted for. How is this influencing the results achieved concerning the shape of the vortices for different propeller speeds?

I know, the weight of the seed particles is probably negligible with respect to the water mass. Probably, this is one reason why the shape of the vortex is not being much affected in the numerical simulations.

Author Response

Dear Reviewer,

We greatly appreciate the Reviewer’s valuable comments and suggestions concerning our paper. Please find below a list of our answers and changes to the manuscript.

Yours faithfully,

The corresponding author

Sergey Komarov

 Although the numerical simulations were quite adequate to help explaining the experiments, the disperse phase (i.e., the penetrating particles) is not accounted for. How is this influencing the results achieved concerning the shape of the vortices for different propeller speeds?

 I know, the weight of the seed particles is probably negligible with respect to the water mass. Probably, this is one reason why the shape of the vortex is not being much affected in the numerical simulations.

Thank you for this valuable comment. Yes, the weight of particles is only 0.2% of that of water, and such a small amount can not affect the fluid dynamics in the water bath. Besides, we performed visual observation of the vortex surface and shape before and after adding the particles to the vortex surface and did not find any significant changes. Therefore, we considered that the particle addition does not affect the vortex behavior. We added a mention about this in the revised manuscript (line 237-238).